# Insights into Gene Regulation under Temozolomide-Promoted Cellular Dormancy and Its Connection to Stemness in Human Glioblastoma

**DOI:** 10.3390/cells12111491

**Published:** 2023-05-27

**Authors:** Carolin Kubelt, Dana Hellmold, Daniela Esser, Hajrullah Ahmeti, Michael Synowitz, Janka Held-Feindt

**Affiliations:** 1Department of Neurosurgery, University Medical Center Schleswig-Holstein UKSH, Campus Kiel, 24105 Kiel, Germany; dana.hellmold@uksh.de (D.H.); hajrullah.ahmeti@uksh.de (H.A.); michael.synowitz@uksh.de (M.S.); 2Institute of Clinical Chemistry, University Medical Center Schleswig-Holstein UKSH, Campus Kiel, 24105 Kiel, Germany

**Keywords:** glioblastoma, temozolomide, dormancy, stemness, Chemokine (C-C motif) Receptor-Like (CCRL)1, Schlafen (SLFN)13, Sloan-Kettering Institute (SKI), Cdk5 and Abl Enzyme Substrate (Cables)1, Dachsous Cadherin-Related (DCHS)1

## Abstract

The aggressive features of glioblastoma (GBM) are associated with dormancy. Our previous transcriptome analysis revealed that several genes were regulated during temozolomide (TMZ)-promoted dormancy in GBM. Focusing on genes involved in cancer progression, Chemokine (C-C motif) Receptor-Like (CCRL)1, Schlafen (SLFN)13, Sloan-Kettering Institute (SKI), Cdk5 and Abl Enzyme Substrate (Cables)1, and Dachsous Cadherin-Related (DCHS)1 were selected for further validation. All showed clear expression and individual regulatory patterns under TMZ-promoted dormancy in human GBM cell lines, patient-derived primary cultures, glioma stem-like cells (GSCs), and human GBM ex vivo samples. All genes exhibited complex co-staining patterns with different stemness markers and with each other, as examined by immunofluorescence staining and underscored by correlation analyses. Neurosphere formation assays revealed higher numbers of spheres during TMZ treatment, and gene set enrichment analysis of transcriptome data revealed significant regulation of several GO terms, including stemness-associated ones, indicating an association between stemness and dormancy with the involvement of SKI. Consistently, inhibition of SKI during TMZ treatment resulted in higher cytotoxicity, proliferation inhibition, and lower neurosphere formation capacity compared to TMZ alone. Overall, our study suggests the involvement of CCRL1, SLFN13, SKI, Cables1, and DCHS1 in TMZ-promoted dormancy and demonstrates their link to stemness, with SKI being particularly important.

## 1. Introduction

Glioblastoma (GBM) represents the most common and most malignant primary brain tumor in adults [1]. Besides its highly invasive nature, its resistance to chemo- and radiotherapy, the inevitable incidence of recurrences, and a vast intra- and intertumoral heterogeneity account for the up-to-now incurability of this tumor type. Intense research and technological advancements have allowed an increasing subclassification of the heterogenous tumor entity [2], even though, to date, no breakthrough in therapy permitting a significant prolongation of life expectancy has been accomplished. Across subtypes, the aggressive properties of GBM were shown to be linked to distinct phenomena such as glioma stem-like cells (GSCs) and dormancy [3]. Since GSCs possess the capacity to self-renew and initiate a tumor, and play a decisive role in tumor progression and relapse, they represent an exciting starting point concerning new therapeutic approaches [4]. In the previous work of our group, we were able to prove striking parallels between stemness and the concept of cellular dormancy in GBM [3]. As cellular dormancy depicts a reversible growth arrest of cells, dormant cells can escape conventional treatment strategies since they mainly affect fast-dividing cells. With time, the dormant state can be abandoned, leading to tumor recurrence following therapy. The entry into dormancy in GBM was shown to be characterized by the upregulation of a specific dormancy-associated gene set [5]. Interestingly, the agent temozolomide (TMZ) itself, used as standard chemotherapy in GBM, was shown to induce entry into a dormant stage [3]. Given this, in the framework of a previously performed microarray-based transcriptome analysis, our group investigated the influence of microenvironmental factors on GBM gene expression during TMZ-promoted cellular dormancy entry and exit. Altogether, 1512 genes were differentially regulated during TMZ-promoted cellular dormancy entry and 1381 during dormancy exit [6]. To narrow down the number of particularly interesting genes for this study, we only selected (1) known genes that (2) were regulated during TMZ-promoted cellular dormancy entry or exit in this specific setup with at least a log2FC = 1.3 value, (3) were expressed to clearly detectable extents after TMZ treatment, and (4) which could also be analyzed at the protein level. After this preselection, we focused on genes already described to be involved in tumor development, the progression or repression of malignancies, and to be connected to the phenomenon of stemness in the broadest sense. Following this procedure, we decided to exemplarily investigate five genes, namely Chemokine (C-C Motif) Receptor-Like (CCRL)1, Schlafen (SLFN)13, Sloan-Kettering Institute (SKI), Cdk5 and Abl Enzyme Substrate (Cables)1, and Dachsous Cadherin-Related (DCHS)1, to further evaluate their significance in GBM. 

CCRL1 is an atypical chemokine receptor that was shown to predominantly exhibit tumor-restricting effects in different malignancies [7,8,9]. However, other studies found the promotion of epithelial-to-mesenchymal transition (EMT) by CCRL1 and hence postulated a tumor-promoting effect [10]. SLFN13 belongs to a family of genes that are involved in cell cycle regulation and mediate growth-inhibitory responses. Its function, especially in cancer, is still poorly understood. An analysis of “The Cancer Genome Atlas” database revealed the downregulation of SLFN13 in breast cancer, lung squamous carcinoma, prostate cancer, and rectal carcinoma, whereas the protein was upregulated in pancreatic- and renal-cell carcinoma [11]. SKI is a proto-oncogene overexpressed in tumor cells of various malignancies and hence involved in the growth, proliferation, invasion, metastasis, and tumor progression of cancer cells [12,13,14]. However, SKI was also shown to express the effect of a tumor suppressor gene in lung cancer [15]. Cables1 is a cyclin-dependent kinase-binding protein that was shown to be involved in the cell cycle, mitosis, cell death, development, and differentiation [16,17]. In multiple types of cancer, a very frequent loss of Cables1 has been observed which implies a potential suppressive effect on tumorigenesis [18]. However, a strong Cables1 expression was found in breast and pancreatic cancers [19]. DCHS1, also known as Cadherin (CDH)19, belongs to the cadherin superfamily and establishes and maintains intercellular connections [20]. It has been attributed to an important role in development, especially in the proliferation and differentiation of neural progenitor cells [20]. In different tumor types, DCHS1 seems to execute a tumor-suppressive effect [21,22,23]. 

To date, only limited-to-no data concerning the role of CCRL1, SLFN13, SKI, Cables1, and DCHS1 in GBM are available, and their relation to phenomena known to be associated with the high therapy resistance of the disease, such as dormancy and stemness, are still mainly uncharted. Hence, this study aimed to further validate the role of the markers in TMZ-promoted cellular dormancy, to examine whether CCRL1, SLFN13, SKI, Cables1, and DCHS1 inherit a potential connection to the phenomenon of stemness in GBM, and investigate whether targeting (any of) these markers can improve the antitumor potential of TMZ. 

## 2. Materials and Methods

### 2.1. Human Specimens

Human tumor samples (*n* = 20) were obtained by surgical dissection at the Department of Neurosurgery (Kiel, Germany) with the approval of the ethics committee of the University of Kiel, Germany, after the written informed consent of donors (file reference: D471/15 and D524/17) and in accordance with the Helsinki Declaration of 1975, revised in 2013. Tumors were diagnosed and classified, according to World Health Organization (WHO) criteria, as GBMs CNS WHO Grade 4 by a pathologist (University Medical Center Hamburg-Eppendorf, UKE, Hamburg, Germany).

### 2.2. Human Glioblastoma (GBM) Cell Lines, Primary Culture Cells, and Stem-like Cells

The human glioblastoma cell lines LN229 (ATCC-CRL-2611), U251 (ECACC 89081403; formerly known as U373MG), U87MG (ECACC 89081402), and T98G (ECACC No. 92090213) were obtained from the European Collection of Authenticated Cell Cultures (ECACC, Salisbury, UK) or the American Type Culture Collection (ATCC, Manassas, VA, USA) and cultured as described previously [24]. Human primary GBM cultures (*n* = 2) were produced by dissociation and cultured according to established techniques as described before [24]. Human primary GBM stem-like cell cultures (*n* = 8) as well as GBM cell line-derived stem-like cells were established and intensively characterized by the formation of neurospheres, the ability to survive and proliferate under stem cell conditions, and the ability to differentiate into more mature cells as described before [3,25,26,27]. The purity of the GBM cells was ascertained by immunostaining with cell type-specific markers and by the absence of contamination with mycoplasms. GBM cell line identity was verified by short tandem repeat profiling at the Department of Forensic Medicine (Kiel, Germany) using the Powerplex HS Genotyping Kit (Promega, Madison, WI, USA) and the 3500 Genetic Analyzer (Thermo Fisher Scientific, Waltham, MA, USA) as previously described [3].

### 2.3. Stimulation of Glioblastoma (GBM) Cells 

As previously described in detail, 1.5  ×  10^5^ LN229, U251, primary culture (PC)a, or PCb cells, respectively, were stimulated for 10 days with TMZ (500 µM, Sigma-Aldrich, St. Louis, MO, USA) dissolved in dimethyl sulfoxide (DMSO, Merck Millipore, Darmstadt, Germany) in Dulbecco’s modified Eagle’s medium (DMEM; Thermo Fisher Scientific) supplemented with 10% fetal bovine serum (FBS; Thermo Fisher Scientific). DMSO 0.5% (*v*/*v*) was used as a control. Hereafter, the medium was changed, and the cells were cultured for another 15 days without TMZ stimulation. Stem-like U251 and LN229 cells were stimulated under the same conditions and for the same periods, but in neurosphere medium [50% DMEM, 50% F12 medium (Thermo Fisher Scientific) containing the following supplements: 2 mM L-glutamine, 0.6% glucose (Roth, Karlsruhe, Germany), 9.5 ng/mL putrescine dihydrochloride (Sigma-Aldrich), 6.3 ng/mL progesterone (Sigma-Aldrich), 5.2 ng/mL sodium selenite (Sigma-Aldrich), 0.025 ng/mL insulin (Sigma-Aldrich), 2 µg/mL heparin (Sigma-Aldrich), and 4 mg/mL bovine serum albumin (Thermo Fisher Scientific). The growth factors EGF (epidermal growth factor; PeproTech, Rocky Hill, NJ, USA) and bFGF (basic fibroblast growth factor; ImmunoTools, Friesoythe, Germany) were added at a concentration of 20 ng/mL as described before [27]. In addition, native LN229 cells were stimulated for 10 days with TMZ (500 µM, Sigma-Aldrich) dissolved in DMSO (Merck Millipore) in DMEM (Thermo Fisher Scientific) supplemented with 10% FBS (Thermo Fisher Scientific) alone or in combination with Disitertide (P144; 100 µg/mL; Tocris Bioscience, Bristol, UK) dissolved in 0.01 M phosphate-buffered saline (PBS), pH 7.4. DMSO 0.5% (*v*/*v*) and PBS were used as controls. Then, the medium was changed and the cells were used for different experiments or cultured for another 11 days without TMZ stimulation but with the continuous addition of Disitertide (100 µg/mL) [6].

### 2.4. Reverse Transcription and Quantitative Real-Time PCR (qRT–PCR)

RNA of cells and tissue were isolated with the TRIzol^®^ reagent (Invitrogen, Carlsbad, CA, USA) or with the ARCTURUS^®^ PicoPure^®^ RNA isolation kit (Applied Biosystems, Foster City, CA, USA) according to the manufacturer’s instructions. DNase digestion, cDNA synthesis, and qRT–PCR were performed as previously described [28] using TaqMan primer probes (Applied Biosystems) listed in Appendix A. Cycles of threshold (C_T_) were determined, and the ∆C_T_ values of each sample were calculated as C_T gene of interest_ − C_T GAPDH_. Either ∆C_T_ values or linearized ∆C_T_ values (2^−∆CT^) are shown in the figures. The regulation of gene expression upon stimulation with Disitertide is displayed as *n*-fold expression changes = 2^∆C^_T_
^control − ∆C^_T_
^stimulus^.

### 2.5. Immunofluorescence Staining 

Cryostat sections of GBM ex vivo tissues were prepared as previously described [3]. Cells were incubated overnight with the primary antibodies at 4 °C, followed by the secondary antibodies for 1 h at 37 °C. The nuclei were counterstained with 4′,6-diamidino-2-phenylindole (Thermo Fisher Scientific; 1:30,000, 30 min, room temperature) and the embedded slides were analyzed by fluorescence microscopy (AxioObserver.Z1; Carl Zeiss AG, Oberkochen, Germany) using the ZEN 3.5 (blue edition) software (Carl Zeiss AG). Used primary antibodies are listed in Appendix A. If primary antibodies were derived from the same species, non-specific binding was blocked by F(ab) fragments derived from that species (1:1000, from Jackson ImmunoResearch, West Grove, PA, USA). Primary antibodies were omitted for negative controls. Donkey anti-mouse or anti-rabbit IgGs labeled with Alexa Fluor 488 or Alexa Fluor 555 (1:1000; Thermo Fisher Scientific) served as secondary antibodies.

### 2.6. Gene Set Enrichment Analysis 

Gene set enrichment analyses (GSEA) were performed with the tool gProfiler based on the gene ontology (GO) source ‘biological process’ [29]. *p*-values were adjusted using a Benjamini–Hochberg FDR correction.

### 2.7. Cytotoxicity Assay and Determination of Proliferation 

The cytotoxic effects were determined using the CytoTox-FluorTM Cytotoxicity Assay (Promega) according to the manufacturer’s instructions and as described before [27]. Supernatants of treated and control cells were collected at days 10 and 21 of stimulation, mixed with the bis-AAF-R110 substrate, and measured in a fluorescence microplate reader (Infinite M200Pro, TECAN, Zürich, Switzerland) at 485/535 nm. The numbers of dead cells were determined according to a prepared standard of digitonin-lysed (82.5 µg/mL; Merck Millipore) cell dilutions. Cell survival/proliferation was determined by counting viable cells with a hemocytometer at days 0, 10, and 21 of the treatment. The percentages [%] of dead cells were calculated as the *n*-fold number of viable cells as described in Equations (1) and (2) after 10 and 21 days of stimulation, respectively. Growth rates were calculated as an *n*-fold number of alive cells compared to day zero of the treatment.
(1)$$\mathrm{Dead}\;\mathrm{cells}\;(\mathrm{day}\;10)\;[\%]=\frac{number\;of\;dead\;cells\;\left[day\;10\right]}{number\;of\;dead\;cells\;\left[day\;10\right]\;+\;vital\;cells\;[day\;10]}\times 100$$
(2)$$\mathrm{Dead}\;\mathrm{cells}\;(\mathrm{day}\;21)\;[\%]=\frac{number\;of\;dead\;cells\;\left[day\;10\;+\;day\;21\right]}{number\;of\;dead\;cells\;\left[day\;10\;+\;day\;21\right]+vital\;cells\;[day\;21]}\times 100$$

### 2.8. Self-Renewal Capacity and Extreme Limiting Dilution Assay 

The self-renewal capacity of 10- versus 3-day-TMZ-pretreated cells and 10 day-TMZ- or TMZ + Disitertide-pretreated cells were measured using an extreme limiting dilution analysis (ELDA) as described before [3]. Briefly, remaining cells after treatment were determined, and decreasing numbers (1600–800–400–200–100–75–50–25–10–5–1 cells per well) of cells were cultured in neurosphere medium (see above), plus 20 ng/mL of bFGF and 20 ng/mL of EGF as described before [3]. Cultures were maintained until day 10 when the number of spheres per well and wells containing spheres for each cell plating density (number of positive cultures) were recorded and plotted using the online ELDA program25 (http://bioinf.wehi.edu.au/software/elda, accessed on 13 April 2023) [30].

### 2.9. Statistical and Correlation Analysis

Depending on the experimental setup, either a two-tailed Student’s *t*-test or a one- or two-way analysis of variance (ANOVA) was performed using the GraphPad Prism 8 software (accessed on 13 April 2023; GraphPad Software, San Diego, CA, USA). The sample sizes and a description of the sample collection, including the number of biological/technical replicates, are described in the figure legends. In general, the data are presented as mean  ±  standard deviation. Correlations were calculated with the Pearson correlation index. Statistical significance is marked with asterisks depending on the *p*-value: * *p*  <  0.05, ** *p*  <  0.01, and *** *p*  <  0.001. 

## 3. Results

### 3.1. Expression and Regulation of Selected Genes under Temozolomide (TMZ)-Promoted Cellular Dormancy in Glioblastoma (GBM) Cell Lines and Patient-Derived Primary Cultures

To evaluate the relevance of CCRL1, SLFN13, SKI, Cables1, and DCHS1 in GBM progression, we first examined their gene expression under TMZ-promoted dormancy entry and exit in different GBM cell lines (LN229 and U251) and primary cultures (PCa and PCb), respectively, using our previously established in vitro model with sole-TMZ stimulation. 

Except for SLFN13, which was not found in PCb, all genes were expressed in the regarded cell lines and primary cultures at different levels. The highest gene expression level amongst all examined cell cultures was found for SKI, whereas CCRL1 and especially SLFN13 exhibited overall a rather low gene expression. The gene expression of DCHS1 and also, though to a lesser extent, Cables1, appeared heterogeneous among the different cell cultures. 

Concerning gene regulation under TMZ-promoted cellular dormancy entry and exit, differences were observed amongst the regarded cells. Overall, a more homogeneous pattern of gene regulation was found, particularly within the primary culture group. Whereas most of the examined genes exhibited an upregulation in dormancy entry and exit in the primary cultures, the cell lines revealed a more complex profile of gene regulation (Figure 1). In detail, in LN229, CCRL1 showed downregulation, albeit only in tendency, both entering and leaving quiescence. Furthermore, a statistically significant upregulation of gene expression was observed for SLFN13 (*p* = 0.017), whereas Cables1 (*p* = 0.028) and DCHS1 (*p* = 0.026) revealed a downregulation in TMZ-promoted dormancy exit. Furthermore, Cables1 (*p* = 0.008) and DCHS1 (*p* = 0.026) showed a significantly higher gene expression after 15 days of stimulation with DMSO in comparison to 10 days of stimulation. Albeit not statistically significant, SKI was found to be slightly upregulated during the entry and downregulated during the exit of TMZ-promoted dormancy; however, high standard deviations were observed. U251 cells revealed the downregulation of CCRL1 (*p* = 0.016) after 15 days of stimulation with TMZ in comparison to 10 days. Contrary to this, SLFN13 (*p* = 0.019) was found to be upregulated after 15 days of stimulation with TMZ. In accordance with LN229, the expression of SKI tended to be downregulated during TMZ-promoted dormancy exit in U251 cells, and a significant downregulation was observed when comparing 10 and 15 days of TMZ stimulation (*p* = 0.049). Whereas Cables1 revealed a downregulation during TMZ-promoted dormancy entry (*p* = 0.004) and a trend of upregulation during exit, DCHS1 was downregulated in both scenarios (*p* entry = 0.011; *p* exit < 0.001). In addition, DCHS1 revealed an upregulation after 15 days of stimulation with DMSO versus 10 days (*p* = 0.016), and a downregulation after 15 days of stimulation with TMZ in comparison to after 10 days of treatment (*p* = 0.005). Concerning the primary cultures, tendencies or even statistically significant upregulations for CCRL1 (PCa: *p* entry = 0.021) and Cables1 (PCa: *p* entry = 0.003; *p* exit < 0.001; PCb: *p* exit < 0.001) during TMZ-promoted dormancy entry and exit were observed. Whereas SLFN13 was found to be upregulated during the entry and exit of TMZ-promoted dormancy in PCa (*p* entry = 0.022; *p* exit < 0.001), no expression of the gene was observed in PCb. However, DCHS1 (*p* < 0.001), which was only detected in a low amount and not significantly regulated in PCa under TMZ-promoted dormancy, revealed an upregulation during dormancy exit in PCb. In PCa, SLFN13 (*p* = 0.007) and Cables1 (*p* < 0.001), and in PCb, Cables1 (*p* = 0.004) and DCHS1 (*p* < 0.001), exhibited upregulation after 15 days of TMZ stimulation in comparison to 10 days of stimulation. Data are presented in Figure 1. 

### 3.2. Expression and Correlation of Selected Genes with Each Other in Patient-Derived Glioblastoma (GBM) Ex Vivo Samples 

Next, we examined the basal gene expression of the selected genes in human GBM ex vivo samples to validate our previous findings. All of the selected genes were detected in the patient’s material at different levels. The highest gene expression level was again found for SKI (average ΔC_T_ = 3.44), followed by DCHS1 (average ΔC_T_ = 4.23). SLFN13 (average ΔC_T_ = 6.87), CCRL1 (average ΔC_T_ = 7.19), and Cables1 (average ΔC_T_ = 7.22) altogether exhibited similar comparatively lower gene expression levels (see Figure 2A). 

To determine potential links between the selected genes, a correlation analysis was performed. All genes revealed positive correlations with each other. Particularly strong correlations were found for CCRL1 and SLFN13 (corr. = 0.91), SLFN13 and Cables1 (corr. = 0.97), and CCRL1 and Cables1 (corr. = 0.85). Medium correlations were detected for SKI and DCHS1 (corr. = 0.73), SKI and Cables1 (corr. = 0.69), SKI and CCRL1 (corr. = 0.68), Cables1 and DCHS1 (corr. = 0.62), and DCHS1 and CCRL1 (corr. = 0.56) (see Figure 2B).

### 3.3. Co-Staining Patterns of Selected Molecules with Each Other in Patient-Derived Glioblastoma (GBM) Ex Vivo Samples

Given the positive correlations found between the selected genes, immunofluorescence double staining of the respective molecules with each other was performed. Since this is a non-quantitative methodology and, in most cases, only individual or small groups of cells exhibit clear co-staining, a purely qualitative assessment of staining was performed here. Overall, staining for all five proteins was detected in the GBM samples. A co-staining of the molecules and solely positive cells was observed in all different staining combinations in varying amounts. Whereas most of the markers revealed either direct co-staining or solely positive cells, SLFN13 and Cables1 also often seemed to be stained in different structures of the same cell. Representative staining examples are shown in Figure 3.

### 3.4. Cellular Sources of Selected Molecules in Patient-Derived Glioblastoma (GBM) Ex Vivo Samples 

To identify the cellular sources of the investigated genes, immunofluorescence double staining of the selected molecules with cell type-specific markers was carried out. Von Willebrand factor (vWF) served as a marker for endothelial cells, a cluster of differentiation molecule (CD)11b tagged microglia, and glial fibrillary acidic protein (GFAP) detected cells of astroglial origin. Furthermore, the stemness markers octamer binding transcription factor (OCT)4, sex-determining region Y-box (Sox)2, and krüppel-like factor (KLF)4 were used to detect a possible link of the markers to tumor stem-like cells. 

CCRL1, SLFN13, SKI, Cables1, and DCHS1 were stained with or nearby vWF to different extents. Whereas SLFN13, SKI, and Cables1 mainly exhibited direct co-staining with vWF, respectively, DCHS1 and especially CCRL1 also seemed to be expressed in different structures of the same vWF-positive cell. Concerning the microglial marker, CD11b, CCRL1, SLFN13, SKI, and DCHS1 seemed to be expressed in different structures of the same CD11b-positive cell. In contrast, Cables1 revealed either a co-staining or was found to be stained separate from CD11b. All of the examined dormancy-associated markers were also found to be stained in GFAP-positive areas. In particular, SKI, Cables1, and DCHS1 revealed a co-staining with GFAP. Interestingly, all of the mentioned markers exhibited individual co-staining patterns with stemness markers. Since tumor stem-like cells are known to represent only a small subpopulation within the total tumor mass (ranging from ~2–20% depending on GBM and stem-like cell subtypes [31]), only single or small groups of double-positive cells have usually been found. Whereas CCRL1, SLFN13, and SKI most frequently appeared directly co-stained with the investigated stemness markers, Cables1 and DCHS1 also often seemed to be stained in different structures of the same cell. Single positive cells for all examined markers were detected. Representative staining examples are presented in Figure 4. 

### 3.5. Correlation Analysis of Dormancy-Associated Genes and Stemness Markers in Patient-Derived Glioblastoma (GBM) Ex Vivo Samples 

Based on the previously described finding of a co-staining for CCRL1, SLFN13, SKI, Cables1, and DCHS1 with stemness markers, respectively, we examined the gene expressions of OCT4, Sox2, and KLF4 in human GBM ex vivo samples. All stemness markers were clearly detected in the samples to different extents (Figure 5). The highest gene expression was found for Sox2 (average ΔC_T_ = 4.37), followed by KLF4 (average ΔC_T_ = 7.0), and OCT4 (average ΔC_T_ = 7.14). To validate the detected link between the genes regulated under TMZ-promoted dormancy and the stemness markers, we performed a correlation analysis. Positive correlations were found, especially for the stemness markers OCT4 and KLF4 with CCRL1, SLFN13, and SKI. In detail, medium correlations were found for OCT4 and SKI (corr. = 0.79), CCRL1 (corr. = 0.74), and SLFN13 (corr. = 0.68); and for KLF4 and CCRL1 (corr. = 0.75), SKI (corr. = 0.74), SLFN13 (corr. = 0.68), and Cables1 (corr. = 0.58). Sox2 exhibited medium correlations with DCHS1 (corr. = 0.65), and CCRL1 (corr. = 0.54). Sox2 and Cables1 (corr. = 0.47), as well as Sox2 and SLFN13 (corr. = 0.45) only revealed weak correlations. 

### 3.6. Expression of Selected Genes in Stem-like Cells Generated from Glioblastoma (GBM) Cell Lines or Patient-Derived Primary Cultures

To further validate the link between CCRL1, SLFN13, SKI, Cables1, and DCHS1 with stemness properties, we examined their expression in stem-like cells of different GBM cell lines and patient-derived primary cultures. 

In most cases, the markers were detectable in the stem-like cells generated from commercial cell lines to different extents. Overall, the highest gene expression in all stem-like cell lines was observed for SKI. The other markers revealed a rather heterogenous pattern between the different stem-like cell lines, which was especially observed for SLFN13 and Cables1. In LN229, the gene expression of SKI (average ΔC_T_ = 7.65) was followed by CCRL1 (average ΔC_T_ = 9.31), DCHS1 (average ΔC_T_ = 10.96), Cables1 (average ΔC_T_ = 12.02), and SLFN13 (average ΔC_T_ = 13.89). In U251, the gene expression of SKI (average ΔC_T_ = 6.05) was followed by DCHS1 (average ΔC_T_ = 8.92), Cables1 (average ΔC_T_ = 10.66), CCRL1 (average ΔC_T_ = 11.02), and SLFN13 (average ΔC_T_ = 11.91). In U87MG, the gene expression of SKI (average ΔC_T_ = 7.38) was followed by CCRL1 (average ΔC_T_ = 10.47), DCHS1 (average ΔC_T_ = 11.18), SLFN13 (average ΔC_T_ = 14.06), and Cables1 (average ΔC_T_ = 16.57). Finally, in T98G, the gene expression of SKI (average ΔC_T_ = 7.45) was followed by CCRL1 (average ΔC_T_ = 10.16), Cables1 (average ΔC_T_ = 10.50), DCHS1 (average ΔC_T_ = 13.14), and SLFN13 (average ΔC_T_ = 15.77). The data are displayed in Figure 6. 

Except for SLFN13 (ΔC_T_ = 11.08), the patient-derived GBM stem-like cells (n = 8) also mostly revealed an expression of the examined markers. DCHS1 (average ΔC_T_ = 7.26) exhibited the highest gene expression among all markers, closely followed by SKI (average ΔC_T_ = 7.44). The lowest gene expressions were observed for Cables1 (average ΔC_T_ = 11.15), and CCRL1 (average ΔC_T_ = 12.00). Data are shown in Figure 7.

### 3.7. Expression and Regulation of Selected Genes under Temozolomide-Promoted Cellular Dormancy in Stem-like Cells and Neurosphere Formation Assay 

To further corroborate our findings, next, we examined the gene expression of CCRL1, SLFN13, SKI, Cables1, and DCHS1 under TMZ-promoted dormancy entry and exit in LN229 and U251 stem-like cells. The data are displayed in Figure 8. 

Interestingly, the results in LN229 and U251 stem-like cells showed similar trends to those obtained for native LN229 and U251, although there were clear differences in some aspects (please compare to Figure 1). For example, in the LN229 and the U251 stem-like cells, similar to the respective native cells, SKI showed the highest expression in both dormancy entry and exit. Similar to the native cells, a statistically significant induction of SKI was observed for both stem-like cell types after 10 days of TMZ stimulation compared to the control (LN229: *p* = 0.015; U251: *p* = 0.0216), which was more pronounced in LN229 stem-like cells. When considering dormancy exit, SKI was slightly downregulated in LN229 stem-like cells, but further induced in U251 stem-like cells, and this was also in contrast to native U251 cells (*p* < 0.0001). CCRL1 and SLFN13 were rather lowly expressed in both stem-like cell types but showed partly significant induction of gene expression compared to the controls, respectively, after 10 days of TMZ stimulation and a further 15 days of recovery (CCRL1, LN229, entry: *p* = 0.0170; CCRL1, U251 entry and exit: *p* = 0.0001; SLFN13, U251, entry: *p* = 0.0004, and exit: *p* < 0.0001). The Cables1 expression level was at a rather low level in LN229 and U251 stem-like cells and was partially significantly induced in both cell types in dormancy entry and exit (U251 entry and exit: *p* = 0.0005). Interestingly, this aspect was not observed in native LN229 and U215 cells. Finally, a strong statistically significant reduction in DCHS1 expression in dormancy exit was observed, particularly in LN229 stem-like cells (*p* < 0.0001), which was consistent with the results observed in native LN229 cells. DCHS1 expression in U251 stem-like cells was more intermediate and was significantly induced in dormancy exit (*p* < 0.0096), whereas a reduction in gene expression was observed in native U251 cells in dormancy exit compared to the control.

To further support these results, we next performed neurosphere formation assays with extreme limiting dilution analysis (ELDA) to investigate sphere formation capacity after pretreatment with TMZ for 3 or 10 days. Here, previous work by our group using native LN229 as an example has shown that LN229 cells pretreated with TMZ for 10 days exhibited a higher self-renewal capacity compared to cells pretreated for a shorter time, yielding sphere formation even at high dilutions [3]. Since these studies were previously performed only with native LN229, we now performed ELDA analysis with patient-derived primary cells (native PCa cells). The data are displayed in Figure 9. Similar to the results observed for LN229 cells, 10 days of pretreatment with TMZ resulted in a higher neurosphere formation capacity of PCa cells in comparison to 3 days of pretreatment (Figure 9A). In accordance with this, induction of the stemness markers OCT4 and KLF4 was more pronounced after 10 days of TMZ pretreatment, whereas Sox2 expression remained unaffected (Figure 9B). 

### 3.8. Gene Set Enrichment Analysis and Inhibition of Sloan-Kettering Institute (SKI) 

As the relationship between the expression of CCRL1, SLFN13, SKI, Cables1, and DCHS1 and the stemness characteristics of GBM cells became increasingly clear based on our results, we next performed gene set enrichment analyses. We used the microarray-based transcriptome datasets previously published by our group, which analyzed the regulation of gene expression during TMZ-promoted entry and exit from cellular dormancy in GBM cells [6]. In detail, up- and downregulated genes comparing the groups of TMZ versus DMSO in both dormancy entry and exit were used for analysis. 

Indeed, the stemness GO term GO:0019827 (stem cell population maintenance) yielded significant results (*p* = 0.019) for the comparison between TMZ versus DMSO in dormancy entry. Genes assigned to this GO term also included SKI. All data from the gene set enrichment analysis are given in Appendix A, and the significantly regulated GO terms of the comparison of TMZ versus DMSO in dormancy entry and exit are visualized in a heatmap in Appendix A.

Since SKI appeared to be particularly important in TMZ-promoted dormancy and its link to stemness, finally, we examined to what extent TMZ application with the simultaneous inhibition of SKI led to increased cytotoxicity and a decreased proliferation of GBM cells compared to TMZ treatment alone. Using our previously established in vitro model, native LN229 cells were stimulated with TMZ alone or in combination with Disitertide for 10 days, after which TMZ was omitted but Disitertide was added for an additional 11 days. Disitertide (also known as P144) itself is a TGF-β inhibitor, which also mediates its efficacy via the downregulation of SKI at both transcriptional and translational levels [32]. The number of dead cells was examined by cytotoxicity assay after 10 and 21 days of treatment, respectively, and the proliferation of cells was also analyzed over the course of treatment. In parallel, we determined the gene expression of SKI in the Disitertide-treated LN229 cells by qRT–PCR. The results are shown in Figure 10. After both 10 and 21 days of Disitertide stimulation, the significant inhibition of SKI gene expression was observed compared to unstimulated controls (10 days: *p* < 0.0002; 21 days: *p* < 0.0001; Figure 10A). Similarly, a significantly increased cytotoxicity of the combination therapy of TMZ + Disitertide compared to TMZ stimulation alone was observed, especially after 21 days of treatment (Figure 10B; ~20% dead cells with TMZ alone, up to ~70% dead cells with TMZ + Disitertide; *p* < 0.0011). In line with this, treatment with TMZ + Disitertide significantly decreased the proliferation of LN229 to a higher extent in comparison to after the administration of TMZ alone (TMZ alone: *p* < 0.0087; TMZ + Disitertide: *p* < 0.0083, compared to control, respectively).

To further examine whether TMZ application with the simultaneous inhibition of SKI affected stemness properties, we performed neurosphere formation assays with ELDA to investigate the sphere formation capacity of native LN229 cells stimulated for 10 days with TMZ alone or with TMZ in combination with Disitertide. The results are shown in Figure 11. Indeed, compared with TMZ treatment alone, the surviving native LN229 cells of the TMZ + Disitertide stimulation showed a lower ability to form neurospheres, which indicated the inhibition of stemness capacity and further supported the higher efficiency of the combination therapy (Figure 11A). In agreement with this, a lower expression of KLF4 was also detected in cells treated for 10 days with TMZ + Disitertide (Figure 11B, *p* = 0.006).

Overall, our study suggests the involvement of CCRL1, SLFN13, SKI, Cables1, and DCHS1 in TMZ-promoted dormancy and demonstrated their link to stemness with SKI being particularly important. 

## 4. Discussion

GBM is the most aggressive primary brain tumor known and still an incurable disease with a medium life expectancy of 15 months despite surgery, and radio- and chemotherapy [1]. One of the reasons for this disastrous prognosis is the high therapy resistance, which has been shown to be linked to distinct phenomena such as, e.g., dormancy [3]. The currently used chemotherapeutic agent in GBM, TMZ, itself does promote dormancy. In the previous work of our group, we identified different genes which were regulated during drug-promoted dormancy entry and exit [6]. Focusing on genes already described to be involved in tumor development, progression, or repression of malignancies, and to be connected to the phenomenon of stemness in the broadest sense, we selected five promising genes associated with TMZ-promoted dormancy in GBM, namely CCRL1, SLFN13, SKI, Cables1, and DCHS1, for further validation. 

We observed differences in the gene regulation patterns of chosen molecules under TMZ-promoted dormancy entry and exit between different GBM cell cultures. Whereas patient-derived GBM primary cultures revealed an upregulation of most of the markers during the entry and exit of dormancy, cell lines exhibited a more heterogeneous gene-regulation pattern.

GBM is known to exhibit a vast molecular intra- and inter-tumoral heterogeneity, which might account for the differences in the gene regulation observed. Intense research and technical advancements have yielded a subclassification of the tumor into either a classical, mesenchymal, and proneural subtype depending on the molecular signature [2]. Furthermore, the O^6^-methylguanine-DNA methyltransferase (MGMT) expression status does significantly affect TMZ response. Silencing of the DNA-repair enzyme MGMT by its promoter methylation abolishes its inhibitory effects against alkylating agents such as TMZ [33]. Additionally, more and more molecular markers are identified, which might affect gene regulation under TMZ-promoted dormancy entry and exit in a complex way [34,35,36], and hence contribute to the heterogeneous picture of gene regulation between the different GBM cells found. Concerning the investigated markers, to date, only CCRL1 was examined regarding its general gene expression in different molecular subtypes of GBM (isocitrate dehydrogenase mutant vs. wildtype, 1p19q codeletion vs. no codeletion). In this specific context, no clear expression changes were found [37]. 

Consistent with the upregulation of most of the markers during the entry into and exit from dormancy observed in the primary cultures, our study revealed a relationship between CCRL1, SLFN13, SKI, Cables1, and DCHS1 as indicated by co-staining and correlation analysis of their gene expressions. To date, no further studies focusing on a connection between these markers exist. Concerning the function of the respective markers, SLFN13, CCRL1, and SKI, especially, were shown to exert tumor-promoting effects in different malignancies. In accordance with the mainly observed upregulation of SLFN13 during TMZ-promoted dormancy exit in our study, previous investigations revealed an increase in the gene expression with progressive glioma grade and hence with incremental aggressive properties [38]. Concerning CCRL1, which was also mainly found to be upregulated during TMZ-promoted dormancy exit in the primary cultures in our study, previous studies documented a reduction in the adherence of cancer cells to each other and to extracellular matrix proteins, and the promotion of EMT by CCRL1 in breast cancer cells [10,39]. However, CCRL1 was also shown to execute opposite effects as an inhibitor of tumor cell proliferation, a reduction in EMT properties, and the tumor cell migration in breast-cancer, hepatocellular, and, nasopharyngeal carcinoma [7,8,9]. As mentioned above and in accordance with the correlations of SKI with CCRL1 and SLFN13 found in our study, SKI was mainly postulated to exert tumor-promoting effects [13,14,15]. Various mechanisms were identified to investigate its function as an influence on Wnt/beta-catenin, phosphatidylinositol 3-kinase/protein kinase B and transforming growth factor (TGF)-ß signaling pathway [40,41,42]. Despite Chen et al. also postulating a tumor-suppressive function in lung cancer [12], in GBM, SKI was shown to negatively regulate the TGF-β signaling pathway, leading to the promotion of tumor progression [43]. Nevertheless, in the specific setup of TMZ-promoted dormancy exit, SKI revealed a reduced gene expression in our study. In accordance with these contradictory findings, the downregulation of SKI by small interfering ribonucleic acid in pancreatic cancer cells resulted in decreased proliferation, whilst, at the same time also increasing EMT, and invasive and metastatic features were observed [43,44]. The mainly observed upregulation of Cables1 and DCHS1 during dormancy entry and exit in the primary cultures, and the correlations found with CCRL1, SLFN13, and SKI seem contradictory considering the tumor-restricting functions of Cables1 and DCHS1, both located on the same chromosome 18q [22,45], as postulated in the literature. Despite Wu et al. postulating an overexpression of Cables1 in breast and pancreatic cancers [19], most of the previous studies documented a very frequent loss of Cables1 in multiple types of cancer which promoted tumor progression [45]. DCHS1 was also proposed as a tumor suppressor gene candidate in gestational and non-gestational choriocarcinomas [22], and was found to be downregulated in colorectal tumors [23]. To date, no data regarding the expression of Cables1 in GBM are publicly available. In our study, Cables1 was clearly detectable in all cell lines, primary cultures, and GBM ex vivo samples. The interferences of all the genes can be documented concerning their molecular mechanisms of action. For instance, SKI, Cables1, and DCHS1 were all shown to be involved in the Wnt signaling pathway [40,45,46].

However, besides tumor cells, endothelial cells and tumor-associated microglia/macrophages were shown to account for the gene expression of the regarded markers. This finding might also contribute to the connection between the markers observed. Supporting our observations, atypical chemokine receptors, such as CCRL1, are known to be involved in adherence to endothelium and the extravasation from blood vessels [47]. Furthermore, CCRL1, SLFN13, SKI, Cables1, and DCHS1 were all found to be expressed in endothelial cells to different extents [48,49,50,51]. A particularly high expression was detected for SKI and DCHS1, whereas CCRL1 and SLFN13 only exhibited a low gene expression in endothelial cells [49]. Concerning the expression of the markers in tumor-associated microglia/macrophages and in accordance with our findings, an expression of SKI, Cables1, and DCHS1 was also described by previous studies [52,53,54,55,56]. Even though data concerning neither an expression of CCRL1 nor SLFN13 in tumor-associated microglia/macrophages are yet publicly available, both markers were previously described to be involved in immunological processes. Whereas CCRL1 controlled intratumor T cell accumulation and activation in a murine mammary cancer cell line [57], SLFN13 was described as an immune-related biomarker that might predict tumor recurrence in lung cancer after curative resection [58]. 

After performing co-staining and correlation analyses of the studied molecules with stemness-associated markers, and realizing expression studies in both stem-like cells generated from commercial GBM cell lines and patient-derived primary cultures, our results showed, in a first using this approach, an association between CCRL1, SLFN13, Cables1, DCHS1, and SKI, and stemness. Indeed, all the molecules studied were clearly expressed in GSCs and exhibited co-staining with OCT4, Sox2, and KLF4 to varying extents, underscoring the results of correlation analyses. Among them, CCRL1, SLFN13, and SKI particularly showed a correlation with the expression of OCT4 and KLF4. When we examined regulation during TMZ-promoted entry and exit from dormancy in GSCs, CCRL1, SLFN13, Cables1, DCHS1, and SKI were regulated in complex patterns, confirming our results from native GBM cell lines and patient-derived primary cultures. When neurosphere formation assays were performed from TMZ-treated native GBM cells, the ability to form neurospheres and the expression of stemness markers indeed increased during treatment. Finally, gene set enrichment analyses indicated the importance of SKI in the phenomenon of stemness in particular. 

Consistent with our findings, Arslan et al. previously reported SLFN family members of SLFN13 to be expressed in GSCs [38]. Additionally, DCHS1 was postulated as a suitable marker and potential therapeutic target for minimally infiltrative GSCs, since it revealed a low expression in developing neuroectodermal tissue, specific upregulation in GSCs, and a potential angiogenic role in tumorigenesis [59]. Concerning SKI, a connection between SKI, OCT4, and Sox2 was also previously described by Song et al. in pancreatic cancer. Here, enhanced SKI expression increased the expression of the pluripotency maintaining markers such as Sox2 and OCT4, and also components of the sonic-hedgehog pathway (Shh), indicating that SKI might be an important factor in maintaining the stemness of pancreatic cancer stem cells through modulating the Shh pathway [60]. Since SKI is involved in the TGF-ß signaling pathway as previously mentioned [42], which in turn is known to support self-renewal of glioma-initiating stem cells [61], the TGF-ß/SKI pathway appears to be of particular importance in GSCs. Indeed, when inhibiting SKI by Disitertide (P144) in our in vitro model of TMZ-promoted dormancy entry and exit, higher cytotoxicity, a stronger inhibition of GBM cell proliferation, and a reduced neurosphere formation capacity along with lower expression of stemness markers were observed. Disitertide is a TGF-β inhibitor peptide, which can decrease proliferation, migration, invasiveness, and tumorigenicity in GBM cells in vitro by a reduction in SMAD2 phosphorylation, the downregulation of SKI and the upregulation of SMAD7 [32]. Thus, in the context of TMZ-promoted entry and exit from dormancy, additional inhibition of transcriptional target genes of the TGF-β pathway, including SKI, appeared to result in a higher antitumor efficacy than TMZ treatment alone. However, because Disitertide, as a TGF-β inhibitor peptide, did not exclusively act on SKI expression, these effects cannot be attributed to the inhibition of SKI alone, although the observed effects underscore the role of this molecule in TMZ-promoted dormancy in GBMs. 

Despite not being explored in GSCs yet, CCRL1 and Cables1 were shown to be connected to stemness. Whereas CCRL1 was found to label mesenchymal subpopulations in an alveolosphere model of mice [62], Cables1 was detected most robustly in embryonic neural tissues in zebrafish and hence postulated to be important for neural differentiation during embryogenesis [17]. In the setting of hematopoiesis, Cables1 was also found to be predominantly expressed in the progenitor cell compartment of bone marrow, hence suggested to be a stemness marker [45].

Although one limitation of our study is the limited amount of cell lines and primary cultures and the small sample sizes included, which prohibits the generalization of the results, our study points to an involvement of CCRL1, SLFN13, Cables1, DCHS1, and particularly SKI in TMZ-promoted dormancy and reveals their connection to the phenomenon of stemness. It seems that the roles of the markers in this disease are complex and also include the tumor microenvironment. However, our study provides basic descriptive research with initial insights into the functions of the selected genes during GBM progression. Further studies are needed to elucidate the detailed impact and function of the selected genes in GBM. 

## 5. Conclusions

GBM still depicts an incurable disease due to phenomena such as dormancy—a reversible growth arrest even promoted by the standard-of-care TMZ itself—and stemness. These complex mechanisms, which contribute to the high therapy resistance of the disease, consist of a large number of downstream factors, whose activities partly overlap and are still not fully understood. In our study, CCRL1, SLFN13, SKI, Cables1, and DCHS1 were all shown to be regulated under TMZ-promoted dormancy and, besides tumor cells, to be expressed by endothelial cells and tumor-associated microglia/macrophages. Moreover, all of the markers, and of particular importance, SKI, were shown to be related to stemness, which highlights the connection between TMZ-promoted dormancy and this phenomenon. Future research is required to investigate the distinct function of CCRL1, SLFN13, SKI, Cables1, and DCHS1 in GBM. Only by understanding the mechanism involved will it be feasible to overcome the enormous therapy resistance and improve the disastrous outcome of GBM patients in the future. 

## Figures and Tables

**Figure 1 cells-12-01491-f001:**
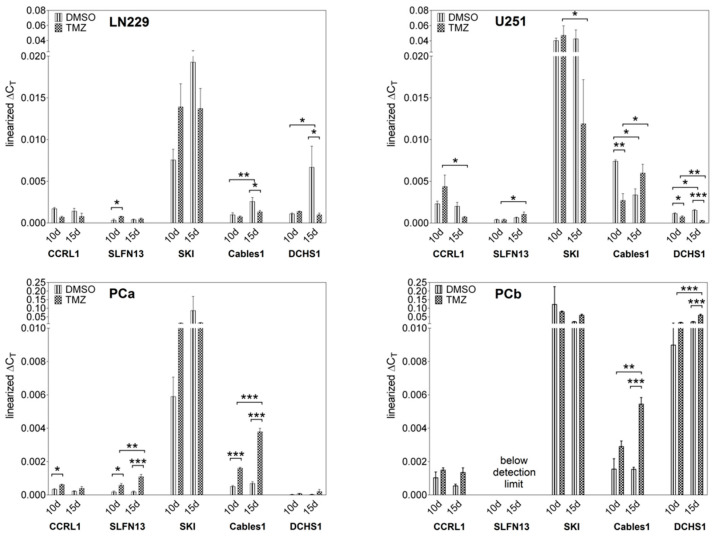
Gene regulation under TMZ-promoted cellular dormancy entry and exit. Gene expression under TMZ-promoted cellular dormancy entry and exit was analyzed by qRT–PCR (*n*  =  3 biological replicates, *n* = 2 technical replicates each). After the treatment of cell lines (LN229, U251) and primary cultures (PCa, PCb) with 500 µM TMZ or 0.5% (*v*/*v*) DMSO, respectively, for 10 days, followed by 15 days without TMZ stimulation, gene expression levels were detected after 10 days of stimulation (dormancy entry) and another 15 days without stimulation (dormancy exit). Gene regulation after TMZ stimulation was statistically analyzed by two-way ANOVA with Bonferroni’s multiple-comparison post hoc test. * *p*  <  0.05, ** *p*  <  0.01, and *** *p*  <  0.001. DMSO: Dimethyl sulfoxide; TMZ: Temozolomide; PCa/b: Primary culture a/b; CCRL1: Chemokine (C-C Motif) Receptor-Like 1; SLFN13: Schlafen 13; SKI: Sloan-Kettering Institute; Cables1: Cdk5 and Abl Enzyme Substrate 1; DCHS1: Dachsous Cadherin-Related 1; qRT–PCR: Reverse transcription and quantitative real-time polymerase chain reaction; ANOVA: analysis of variance.

**Figure 2 cells-12-01491-f002:**
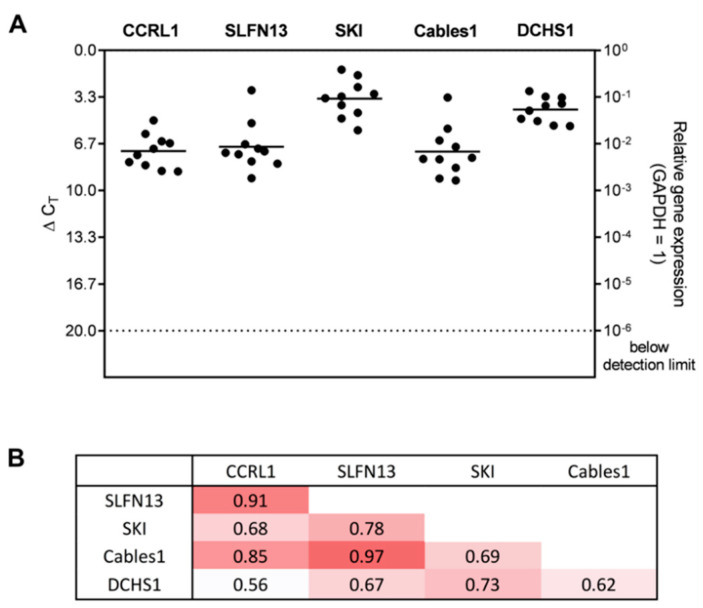
Gene expression of the selected genes in GBM ex vivo samples and their correlation with each other. (**A**) Basic gene expression levels were detected in human GBM ex vivo samples (*n* = 10; *n*  =  2 technical replicates each) by qRT–PCR. Lines represent the mean gene expression for each gene (ΔC_T_ 3.3 = 10-fold expression difference). (**B**) The correlation of gene expression was analyzed by the Pearson correlation index. A darker shade of red corresponds to a higher correlation value. CCRL1: Chemokine (C-C Motif) Receptor-Like 1; SLFN13: Schlafen 13; SKI: Sloan-Kettering Institute; Cables1: Cdk5 and Abl Enzyme Substrate 1; DCHS1: Dachsous Cadherin-Related 1; GAPDH: Glyceraldehyde-3-phosphate dehydrogenase; GBM: Glioblastoma; qRT–PCR: Reverse transcription and quantitative real-time polymerase chain reaction.

**Figure 3 cells-12-01491-f003:**
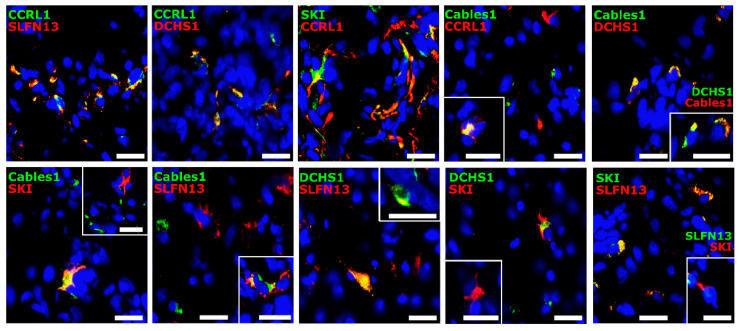
Immunofluorescence double-staining of the selected molecules. Human GBM ex vivo sections (*n*  =  5, different patients; *n*  =  1, technical replicate for each patient) were immunofluorescently stained regarding the presence of co-staining (yellow) for CCRL1, SLFN13, SKI, Cables1, and DCHS1 (green and red, respectively). Nuclei appear blue. Magnification 400×; white bar = 20 µm. CCRL1: Chemokine (C-C Motif) Receptor-Like 1; SLFN13: Schlafen 13; SKI: Sloan-Kettering Institute; Cables1: Cdk5 and Abl Enzyme Substrate 1; DCHS1: Dachsous Cadherin-Related 1; GBM: Glioblastoma.

**Figure 4 cells-12-01491-f004:**
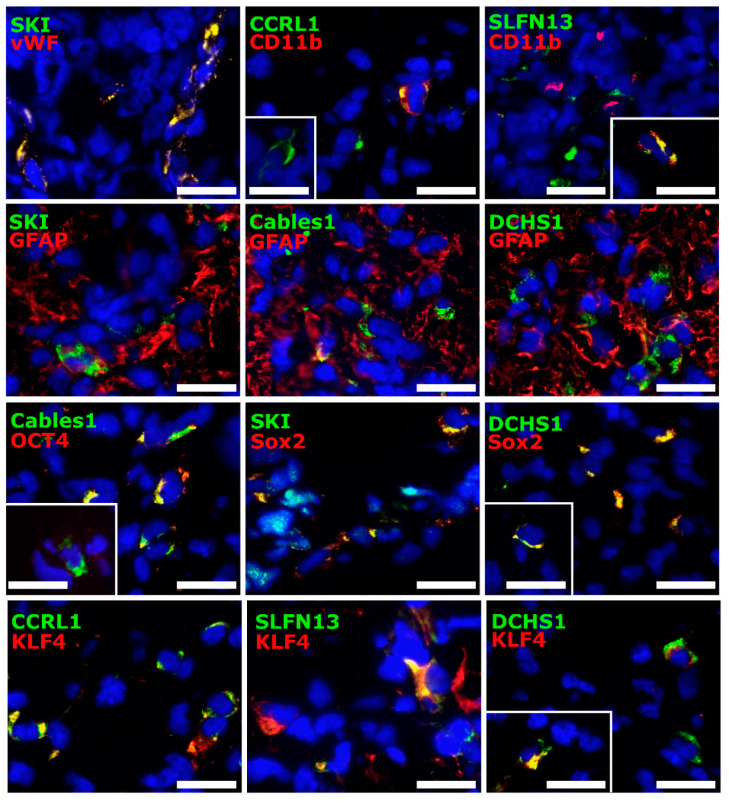
Source of molecules regulated under TMZ-promoted dormancy. Human GBM ex vivo sections (*n*  =  5, different patients; *n*  =  1, technical replicate for each patient) were immunofluorescently stained regarding the presence of a co-staining (yellow) for CCRL1, SLFN13, SKI, Cables1, and DCHS1 (green) with the cell type-specific markers vWF, CD11b, and GFAP and the stemness markers OCT4, Sox2, and KLF4 (red). Nuclei appear blue. Magnification 400×; white bar = 20 µm. CCRL1: Chemokine (C-C Motif) Receptor-Like 1; SLFN13: Schlafen 13; SKI: Sloan-Kettering Institute; Cables1: Cdk5 and Abl Enzyme Substrate 1; DCHS1: Dachsous Cadherin-Related 1; vWF: Von Willebrand factor; CD11b: Cluster of differentiation molecule 11b; GFAP: Glial fibrillary acidic protein; OCT4: Octamer binding transcription factor 4; Sox2: Sex determining region Y-box 2; KLF4: Krüppel-like factor 4; TMZ: Temozolomide; GBM: Glioblastoma.

**Figure 5 cells-12-01491-f005:**
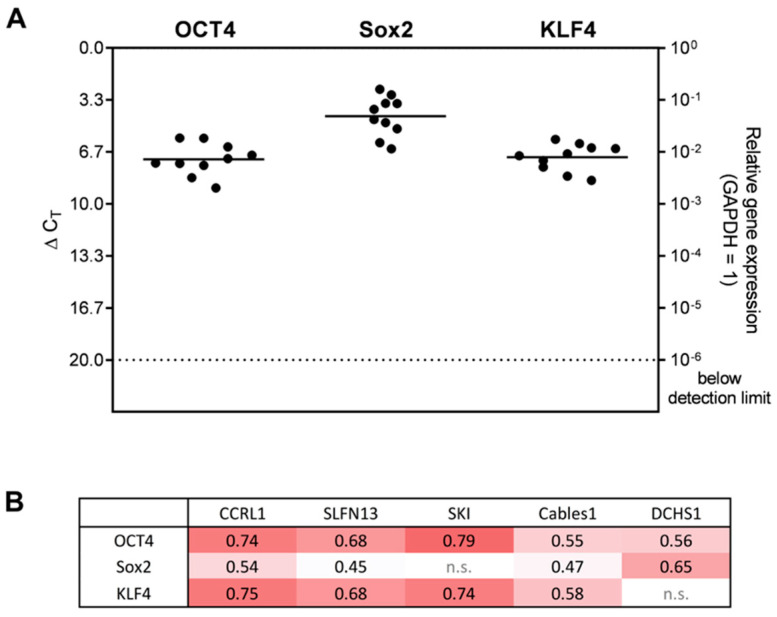
Gene expression of stemness markers in GBM ex vivo samples and their correlation with genes regulated under TMZ-promoted dormancy. (**A**) Basic gene expression levels were detected in human GBM ex vivo samples (*n* = 10; *n*  =  2, technical replicates each) by qRT–PCR. Lines represent the mean gene expression for each gene (ΔC_T_ 3.3 = 10-fold expression difference). (**B**) The correlation of gene expression was analyzed by the Pearson correlation index. A darker shade of red corresponds to a higher correlation value. Non-statistically significant correlations are marked by n.s.. OCT4: Octamer binding transcription factor 4; Sox2: Sex determining region Y-box 2; KLF4: Krüppel-like factor 4; GAPDH: Glyceraldehyde-3-phosphate dehydrogenase; CCRL1: Chemokine (C-C Motif) Receptor-Like 1; SLFN13: Schlafen 13; SKI: Sloan-Kettering Institute; Cables1: Cdk5 and Abl Enzyme Substrate 1; DCHS1: Dachsous Cadherin-Related 1; GBM: Glioblastoma; TMZ: Temozolomide; qRT–PCR: Reverse transcription and quantitative real-time polymerase chain reaction.

**Figure 6 cells-12-01491-f006:**
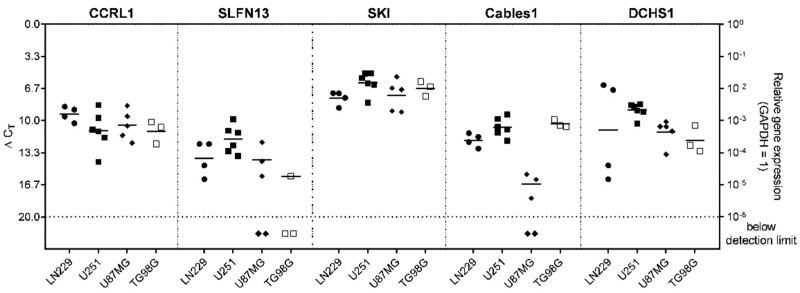
Expression of genes regulated under TMZ-promoted dormancy in stem-like cells generated from commercial cell lines. Basic gene expression levels were detected in stem-like cells from LN229, U251, U87MG, and T98G GBM cells (*n* = 3–6; *n*  =  2, technical replicates each) by qRT–PCR. Lines represent the mean gene expression for each gene, the symbol tags the respective cell line (ΔC_T_ 3.3 = 10-fold expression difference). CCRL1: Chemokine (C-C Motif) Receptor-Like 1; SLFN13: Schlafen 13; SKI: Sloan-Kettering Institute; Cables1: Cdk5 and Abl Enzyme Substrate 1; DCHS1: Dachsous Cadherin-Related 1; GAPDH: Glyceraldehyde-3-phosphate dehydrogenase; TMZ: Temozolomide; qRT–PCR: Reverse transcription and quantitative real-time polymerase chain reaction.

**Figure 7 cells-12-01491-f007:**
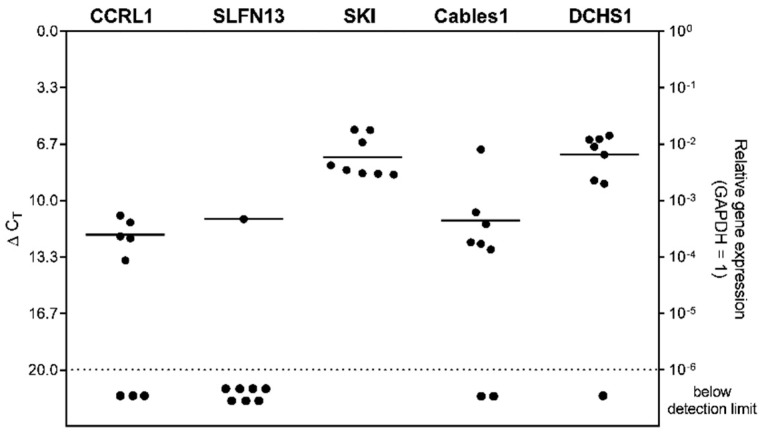
Expression of genes regulated under TMZ-promoted dormancy in patient-derived stem-like cells. Basic gene expression levels were detected in patient-derived GBM stem-like cells (*n* = 8; *n * =  2, technical replicates each) by qRT–PCR. Lines represent the mean gene expression for each gene (ΔC_T_ 3.3 = 10-fold expression difference). CCRL1: Chemokine (C-C Motif) Receptor-Like 1; SLFN13: Schlafen 13; SKI: Sloan-Kettering Institute; Cables1: Cdk5 and Abl Enzyme Substrate 1; DCHS1: Dachsous Cadherin-Related 1; GAPDH: Glyceraldehyde-3-phosphate dehydrogenase; TMZ: Temozolomide; qRT–PCR: Reverse transcription and quantitative real-time polymerase chain reaction.

**Figure 8 cells-12-01491-f008:**
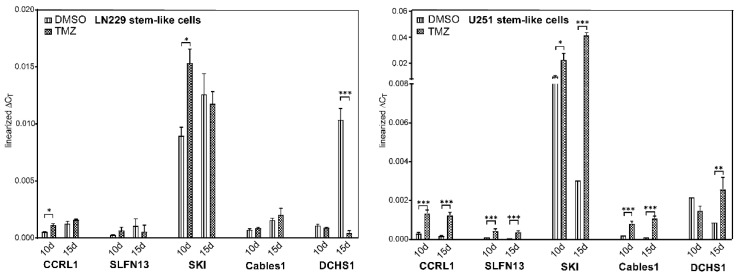
Gene regulation under TMZ-promoted cellular dormancy entry and exit in stem-like cells. Gene expression under TMZ-promoted cellular dormancy entry and exit was analyzed by qRT–PCR (*n*  =  3, biological replicates; *n* = 2, technical replicates each). After the treatment of LN229 and U251 stem-like cells with 500 µM TMZ or 0.5% (*v*/*v*) DMSO, respectively, for 10 days followed by 15 days without TMZ stimulation, gene expression levels were detected after 10 days of stimulation (dormancy entry) and another 15 days without stimulation (dormancy exit). Gene regulation after TMZ stimulation was statistically analyzed by two-way ANOVA with Bonferroni’s multiple-comparison post hoc test. * *p*  <  0.05, ** *p*  <  0.01, and *** *p*  <  0.001. DMSO: Dimethyl sulfoxide; TMZ: Temozolomide; CCRL1: Chemokine (C-C Motif) Receptor-Like 1; SLFN13: Schlafen 13; SKI: Sloan-Kettering Institute; Cables1: Cdk5 and Abl Enzyme Substrate 1; DCHS1: Dachsous Cadherin-Related 1; qRT–PCR: Reverse transcription and quantitative real-time polymerase chain reaction; ANOVA: analysis of variance.

**Figure 9 cells-12-01491-f009:**
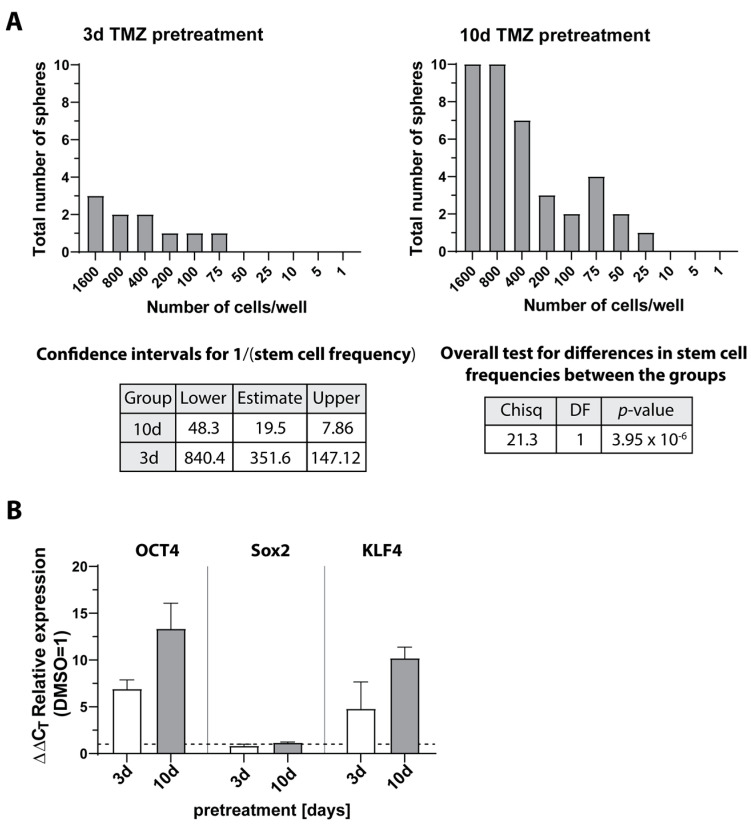
Self-renewal capacity, ELDA, and qRT–PCR analysis. Native primary culture (PCa) cells were stimulated with 500 μM TMZ for 10 and 3 days, and (**A**) self-renewal capacity was determined under stem cell culture conditions with ELDA (*n* = 2). Briefly, cells were plated in decreasing numbers from 1600 cells/well to 1 cell/well. Cultures were maintained until day 10 when the number of spheres per well and wells containing spheres for each cell plating density (number of positive cultures) were recorded and plotted using online ELDA program25; http://bioinf.wehi.edu.au/software/elda, accessed on 13 April 2023; (**B**) expression of stemness markers OCT4, Sox2, and KLF4 was determined by qRT–PCR. The induction of gene expression upon stimulation with TMZ was displayed as n-fold expression changes = 2^∆CT control − ∆CT stimulus^. Error bars correspond to the standard deviation. TMZ: Temozolomide; DMSO: Dimethyl sulfoxide; OCT4: Octamer binding transcription factor 4; Sox2: Sex determining region Y-box 2; KLF4: Krüppel-like factor 4; ELDA: extreme limiting dilution analysis; qRT–PCR: Reverse transcription and quantitative real-time polymerase chain reaction.

**Figure 10 cells-12-01491-f010:**
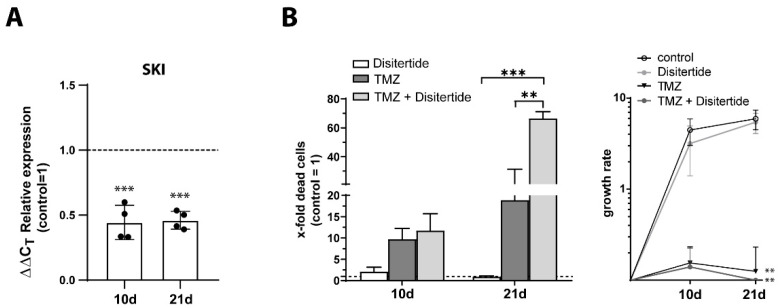
Cytotoxic and antiproliferative effect of TMZ application with simultaneous inhibition of SKI in GBM cells. LN229 cells were treated with TMZ + Disitertide (500 µM TMZ, 100 µg/mL Disitertide) for 10 days, followed by another 11 days without TMZ stimulation but with continuous addition of Disitertide (100 µg/mL). (**A**) Gene expression of SKI was quantified by qRT–PCR at different time points of treatment. The induction of gene expression upon stimulation with Disitertide was displayed as *n*-fold expression changes = 2^∆CT control − ∆CT stimulus^. (**B**) Death rates were obtained by performing a cytotoxicity assay after 10 and 21 days of stimulation, respectively. The cell survival/proliferation was determined by counting viable cells at days 0, 10, and 21 of the treatment. The percentages [%] of dead cells were calculated as the *n*-fold number of viable cells. *n* = 2, biological replicates, with *n* = 2, technical replicates each. The significances between different stimulations were determined using either a two-tailed Student’s *t*-test (**A**) or a two-way ANOVA test followed by a Tukey´s multiple-comparison test (**B**) (** *p* < 0.01; *** *p* < 0.001). Error bars correspond to the standard deviation. TMZ: Temozolomide; SKI: Sloan-Kettering Institute; GBM: Glioblastoma; qRT–PCR: Reverse transcription and quantitative real-time polymerase chain reaction; ANOVA: analysis of variance.

**Figure 11 cells-12-01491-f011:**
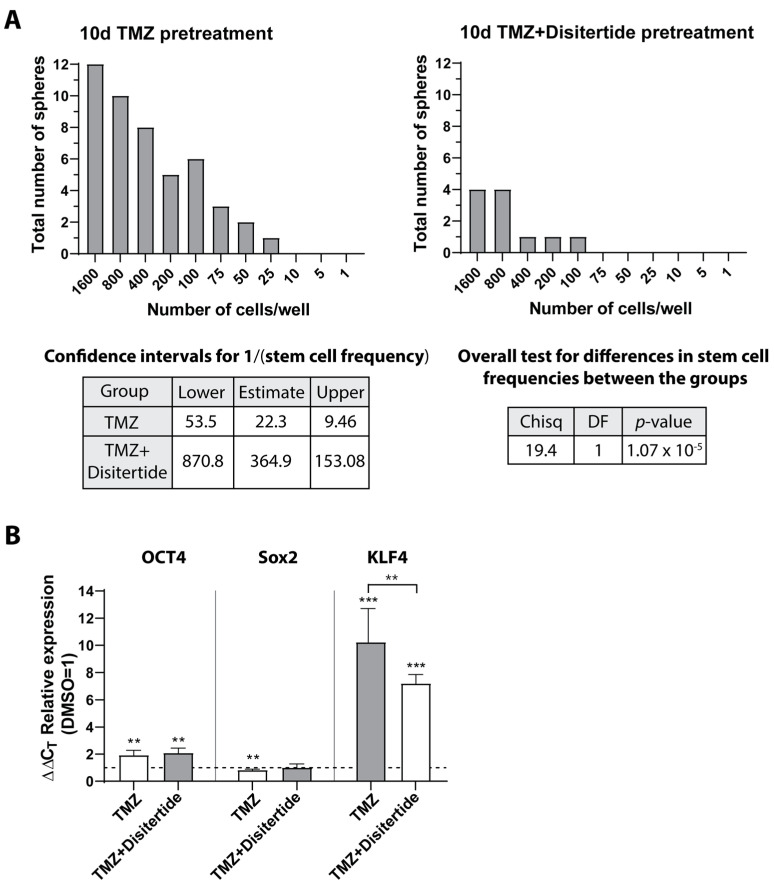
Self-renewal capacity, ELDA, and qRT–PCR analysis. Native LN229 cells were stimulated with 500 μM TMZ or TMZ + Disitertide (500 µM TMZ, 100 µg/mL Disitertide) for 10 days, and (**A**) self-renewal capacity was determined under stem cell culture conditions with ELDA (*n* = 2). Briefly, cells were plated in decreasing numbers from 1600 cells/well to 1 cell/well. Cultures were maintained until day 10 when the number of spheres per well and wells containing spheres for each cell plating density (number of positive cultures) were recorded and plotted using online ELDA program25; http://bioinf.wehi.edu.au/software/elda, accessed on 13 April 2023; (**B**) expression of stemness markers OCT4, Sox2, and KLF4 was determined by qRT–PCR. The induction of gene expression upon stimulation was displayed as n-fold expression changes = 2^∆CT control − ∆CT stimulus^. The significances between different stimulations were determined using either a two-tailed Student’s *t*-test or a one-way ANOVA test followed by a Tukey´s multiple-comparison test (** *p* < 0.01; *** *p* < 0.001). Error bars correspond to the standard deviation. TMZ: Temozolomide; DMSO: Dimethyl sulfoxide; OCT4: Octamer binding transcription factor 4; Sox2: Sex determining region Y-box 2; KLF4: Krüppel-like factor 4; ELDA: extreme limiting dilution analysis; qRT–PCR: Reverse transcription and quantitative real-time polymerase chain reaction; ANOVA: analysis of variance.

## Data Availability

Not applicable.

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
