# Peer review of "Insights into Gene Regulation under Temozolomide-Promoted Cellular Dormancy and Its Connection to Stemness in Human Glioblastoma"

_cells, 2023, doi:10.3390/cells12111491_

Round 1
Reviewer 1 Report (Previous Reviewer 1)
The manuscript by Kubelt et al. describes regulation of five genes under TMZ treatment in GBM cells. Although the initial version of manuscript was too premature to be published, the authors have improved their manuscript according to the reviewer comments. However, I still recommend two things to improve the quality of this article.
1. Please provide the visualized and summarized results of gene set enrichment analysis (e.g., heat map, …).
2. The title of Figure 10 is missing.
N/A
Author Response
Please see the attachment

Reviewer 2 Report (Previous Reviewer 2)
The revision has adequately addressed my concerns.
Author Response
Please see the attachment

This manuscript is a resubmission of an earlier submission. The following is a list of the peer review reports and author responses from that submission.
Round 1
Reviewer 1 Report
The manuscript by Kubelt et al. describes regulation of five genes under TMZ treatment in GBM cells. Although the aim is interesting, this study is too premature to be published. Several issues below should be considered to improve the quality of this article.
- The authors selected five genes to explore, solely based on previous research. What is the rationale of utilizing small subset of transcriptome? The authors need to clearly demonstrate the reason why they selected these genes instead of total transcriptome.
- The authors performed only expression-associated experiments. To persist involvement of dormancy and stemness, functional studies, including gene on-off study, neurosphere formation assay, proliferation assay, should be performed. The presentation of the data is not sufficiently detailed to evaluate the strength of the conclusions.
- To persist involvement of certain phenotypes, such as stemness, enrichment analysis (like over-representation analysis) using phenotype-associated gene sets should be performed, instead of enumerating the expression level of individual genes.
Reviewer 2 Report
Based on author’s previous transcriptome analysis in LN229 cells stimulated by TMZ, 5 genes were selected. Authors further investigated their expression under TMZ-promoted dormancy in human GBM cell lines, patient-derived primary cultures and human GBM ex vivo samples. A correlation analysis was also performed. These observations are interested. However, the conclusions are overstated or not supported.
- It is not clear how the 5 genes were selected. In addition, since the 5 genes were selected from LN229 cells stimulated by TMZ and upregulated, herein the data in Fig 1 are not consistent.
- The size of immunofluorescence images is too small. It is hard to identify their co-staining. In addition, how many positive cells were detected?
- It is important to determine the gene expression difference with or without TMZ stimulation in stem-like cells generated from GBM cell lines or patient-313 derived primary cultures.
- It is important to determine the gene expression difference under TMZ-promoted cellular dormancy entry and exit in stem-like cells generated from GBM cell lines or patient-313 derived primary cultures.